

# Diffuse idiopathic skeletal hyperostosis was the specific risk factors of methicillin-susceptible *Staphylococcus aureus* spine infection: a retrospective study in a single center

Kaori Endo

Sapporo Tokusyukai Hospital, Sapporo, Hokkaido, Japan
National Coalition of Independent Scholars, Battleboro, VT, United States of America

Corresponding author
Kaori Endo,
kaoriendo.2022@gmail.com

## ABSTRACT

**Aim**. To investigate how risk factors and reduced spinal mobility contribute to spinal infections arising from methicillin-susceptible *Staphylococcus aureus* (MSSA) bacteremia, known for increased mortality and diagnostic difficulties, especially in patients with septic shock or coma.

**Methods**. This retrospective study divided MSSA bacteremia patients into three groups: spinal infections (Group A, $n = 14$), non-spinal/implant infections (Group B, $n = 24$), and implant-related infections (Group C, $n = 21$). Analyses focused on demographics, medical history, laboratory inflammatory markers at antibiotic initiation, and spinal pathologies detected by CT. All results of the statistical analyses were significant at $P < 0.05$. We employed multinomial univariable logistic regression and contingency table analysis to assess risk factors across three groups. Subsequently, binomial multivariable logistic regression was used to compare Group A against Groups B and C, successfully identifying significant predictors of spinal infection.

**Results**. A lower incidence of diabetes ($p = 0.029$), higher C-reactive protein (CRP) levels at onset ($p = 0.014$), and the presence of diffuse idiopathic skeletal hyperostosis (diffuse idiopathic skeletal hyperostosis (DISH); $p = 0.022$) were significantly associated with spinal infections in Group A. Furthermore, binomial analysis revealed DISH (Odds Ratio (OR) = 41.750; 95% Confidence Interval (CI) [1.86–939.0]; $p = 0.019$), absence of diabetes (OR = 1.20, CI [1.01–1.43], $p = 0.038$), elevated CRP (OR = 23.34, CI [1.13–483.4], $p = 0.042$), and a lower day 3/day 1 white blood cell (WBC) ratio (OR = 0.964, CI [0.93–1.00], $p = 0.047$) as risk factors when compared with other groups.

**Conclusion**. Spinal infection patients with MSSA bacteremia are less likely to have diabetes and more likely to have higher initial CRP levels and DISH. Notably, DISH might be emerging as a distinctive risk factor for spinal infection, underscoring its potential as a marker for clinical awareness.

## INTRODUCTION

Spinal infections such as discitis, osteomyelitis, and facet infections lead to high mortality and catastrophic complications (*Berbari et al., 2015*). An increasing incidence of spinal infections has been observed, particularly among the elderly and those with immunodeficiencies (*Devico Marciano et al., 2023*; *Yamada et al., 2021*). Infections are typically diagnosed in the setting of strong back pain unresponsive to conservative treatments and elevated inflammatory markers, and then the diagnosis is confirmed with a blood culture or biopsy and magnetic resonance imaging (*An & Seldomridge, 2006*). However, a correct diagnosis is exceedingly challenging in patients with altered consciousness due to septic shock.

Methicillin-susceptible *Staphylococcus aureus* (MSSA) is the predominant pathogen that causes spinal infections, and blood cultures positive for MSSA are considered clinically significant and should prompt immediate clinical assessment and initiation of empirical therapy (*Berbari et al., 2015*). Delays in source control, rather than the specifics of antistaphylococcal therapy, have been significantly associated with extended bacteremia duration and worse outcomes, including metastatic complications, length of hospital stay, and 30-day mortality (*Minejima et al., 2020*). All patients with MSSA bacteremia should undergo echocardiography or transthoracic echocardiography if they have a VIRSTA score ≥3, and the evaluation should be tailored to findings based on history and physical examination (*Heriot et al., 2020*; *Peinado-Acevedo et al., 2021*).

However, the lack of a risk-scoring system for the definitive diagnosis of spinal infections with MSSA bacteremia exacerbates the diagnostic challenges. Advanced imaging techniques, including MRI, gallium-67 single-photon emission computed tomography, radionuclide studies, and positron emission tomography, have effectively demonstrated that infections of the disc, spinal body, and facet joints are primary infectious entities of the spine, potentially associated with or complicated by the formation of epidural or paraspinal muscle abscesses (*An & Seldomridge, 2006*; *Berrevoets et al., 2017*; *Ghanem-Zoubi et al., 2021*). If such imaging tests are not sufficient or confirmed, early imaging or computed tomography (CT)-guided drainage (*Diffre et al., 2020*; *Lee et al., 2020*), posterolateral percutaneous full-endoscopic debridement and irrigation (*Ito et al., 2007*; *Ito et al., 2009*; *Yamada et al., 2022*), herniotomy, and debridement with medial facetectomy are additionally required by collecting a sample directly from the infected disc, endplate, or vertebral body. Although these invasive procedures can improve outcomes, they are not without risks and are less feasible in emergency situations or severe cases.

Plain CT does not directly diagnose spinal infection; nevertheless, it is more informative regarding calcification changes (*An & Seldomridge, 2006*). The literature shows that the more degenerative and unstable the spinal area, the higher the incidence of spinal infection (*Muffoletto et al., 2001*). Degenerative changes caused by instability due to malalignment (*e.g.*, idiopathic or degenerative scoliosis), non-union or collapse of the vertebral body, and adjacent segmental diseases following strong bony fusion (such as bridging, ligament ossification, and particularly diffuse idiopathic skeletal hyperostosis (DISH)) are the focus of this study. Calcification of local spine unions highly affects the other spine, discoid, and
facet stresses, leading to more degenerative changes. Our hypothesis posits that factors limiting range of motion, including scoliosis, bridging, and DISH, may also contribute to the risk of spinal infection. Moreover, we hypothesized that areas with the highest Hounsfield unit (HU) values, corresponding to regions without stability, are likely sites of infection.

The purpose of this study is to investigate how risk factors and reduced spinal mobility contribute to spinal infections and to assess the relative importance of various risk factors for these infections compared to those affecting other organs or associated with surgical implants.

## MATERIALS & METHODS

### Subjects

This study is a retrospective clinical study of MSSA bacteremia patients diagnosed with the two sets of blood cultures and treated at our general hospital from 1st April 2015 to 31st August 2019. Key elements of this study protocol were presented in Fig. 1. The study protocol and opt-out informed consent process were approved by the ethics committee at each site (Tokushukai group IRB TGE 01322-010). Given the retrospective nature of this study, informed consent was waived by the ethics committee, and an opt-out consent approach was implemented to allow participants to decline participation. This study was conducted in compliance with the study plan, local laws, and the Declaration of Helsinki.

Patients were eligible for the study if they were diagnosed with MSSA bacteremia, confirmed by two positive sets of blood cultures within the study period. Patients were excluded if the culture results revealed infections caused by bacteria other than MSSA. Specifically, patients with methicillin-resistant *Staphylococcus aureus* (MRSA) were excluded due to the variable effects of antibiotic treatments on this group, which could confound the study's outcomes. Additionally, patients who had undergone invasive spinal procedures or received medical injections within one month prior to enrollment were excluded.

Patients were classified into three groups based on their infection types: Group A comprised 14 patients with primary spine infections, including nine cases of discitis and osteomyelitis and five cases of facet infections. Group B included 24 patients with other organ infections such as heart, aorta, lung, skin, and muscle infections, and some without a specified infection site. Group C consisted of 21 patients with implant-related infections, specifically infections associated with pacemakers and venous lines. This categorization was based on the distinct diagnostic and treatment approaches characteristic of implant infections, which are typically more straightforward to diagnose and manage compared to the more complex presentations in the other groups.

### Diagnostic procedures

The identification of infection sites was primarily conducted using enhanced CT scans, echocardiography, and spinal MRI, avoiding invasive tests that could alter laboratory inflammation data. Exceptions were made for patients without recent invasive spinal procedures or medical injections within the last month. Antibiotic treatment

MSSA bacteremia patients divided into 3 groups.
A: Spine infection groups (n=14)
B: The other infection without spine and implant (n=24)
C: Implant infection (n=21).

↓

Variables to identify the risk factors
1, Patient factors
   Age, Sex, Past history of diabetes, Past use of steroid
2, Laboratory data
   WBC, CRP on day1
   Antibiotics treatment effect (rate of day3/day1)
3, Spinal changes with CT
   Scoliosis, Bridging, DISH

↓

1, Multinomial univariable logistic regression and contingency table analysis in three groups
To investigate each group's features and risk factors

2, Binominal multivariable logistic regression (objective variables: A vs B and C)
To see the odds ratio of spine infection against the other two groups

**Figure 1** **Flow chart of analytical procedure.** MSSA bacteremia patients were divided into three groups and statistical analysis was conducted. Next, the only spine infection group was assessed qualitatively with CT-MIP.

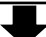

was administered without restriction, following dosages recommended for infectious endocarditis from initial examination until culture results were obtained (*Nakatani et al., 2019*). Surgical treatments were not performed for Groups A and B except for one patient in Group A treated within the first three days. Group C patients underwent device removal within this period. Diagnostic decisions were made collaboratively by physicians, radiologists, and orthopedic surgeons following the Infectious Diseases Society of America (IDSA) guidelines (*Berbari et al., 2015*).

## Data analysis

Patient factors analyzed included age (mean 74.3 ± 18.1 years), gender (male: 23, female: 36), history of diabetes, steroid use, initial white blood cell count (WBC), and C-reactive protein (CRP) levels at the start of antibiotic treatment (Day 1). The effectiveness of the antibiotic treatment was assessed by the rates of WBC and CRP changes from Day 1 to Day 3 (Day 1/3). Additional assessments included the presence of scoliosis (Fig. 2A), bony bridging (Fig. 2B), and DISH, with details referred to in the introductory section. We also

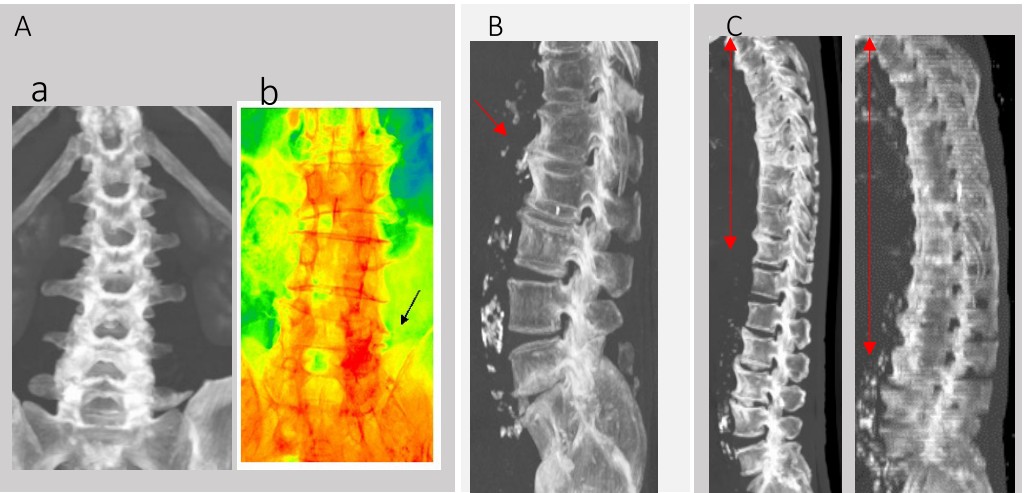

**Figure 2  CT-MIP analysis of spine.** (A) (a) Scoliosis in CT-MIP. (b) CT-MIP color mapping. (c) The black arrow area shows the highest CT value area has degenerative and infection area. (B) Bony bridging. (C) Red area are DISH.

conducted additional checks that this area had been definitively infected with retrospective MRI tests after every 2 weeks.

An additional question investigated was whether the infection sites in Group A ($n = 14$) could be identified through degenerative changes in the spine early in the course of discitis and facet joint infections (Fig. 2Ab). The areas displayed in red had the highest Hounsfield Unit (HU) values in 2D CT-maximum intensity projection (CT-MIP) color imaging, suggesting more pronounced degenerative changes compared to other areas from the reference method (*Zou et al., 2019*).

## Statistical analysis

All results of statistical analyses were significant at $P < 0.05$ using the software JMP Pro (Software Version 17.0.0, SAS Institute Inc., Cary, NC, USA). First, statistical analyses were performed using the multinomial univariable logistic regression and contingency table analysis in the three groups to investigate each group's features and risk factors. Second, a binominal multivariable logistic regression (objective variables: A *vs* B and C) was conducted to see the odds ratio of spine infection against the other two groups.

## RESULTS

### Clinical characteristics of MSSA bacteremia patients and comparison using multinomial univariable logistic regression and contingency table analysis

*Patient characteristics and initial observations*

The results from the multinomial univariable logistic regression and contingency table analysis are detailed in Table 1, and all patient data are included in Supplemental Information 1. The prevalence of diabetes among spine infection patients (Group A)

**Table 1 Clinical characteristics of MSSA bacteremia patients.** The data showed average ± standard deviation. The objective variables are A, B and C.

| Group | A | B | C | All | p-value |
|---|---|---|---|---|---|
| Infection sites | Spine | Other | Implant | | |
| Patient number | 14 | 24 | 21 | 59 | – |
| 1, Multinominal univariable logistic regression | | | | | |
| Age | 69.4 ± 19.3 | 73.5 ± 18.2 | 78.4 ± 17.1 | 74.3 ± 18.1 | 0.317 |
| WBC at day1 (×100/μl) | 172.9 ± 90.2 | 143.1 ± 66.4 | 127.3 ± 53.5 | 144.5 ± 7.0 | 0.160 |
| CRP at day1 (mg/dl) | 18.1 ± 9.1* | 15.9 ± 10.3 | 10.0 ± 6.6 | 14.3 ± 9.3 | 0.014 |
| WBC at day3/day1 (%) | 74.2 ± 19.4 | 104.3 ± 59.2 | 84.0 ± 28.0 | 89.9 ± 43.6 | 0.057 |
| CRP at day3/day1 (%) | 117.3 ± 86.5 | 106.1 ± 61.6 | 141.6 ± 93.9 | 121.4 ± 80.4 | 0.323 |
| 2, Contingency table analysis | | | | | |
| Male (%) | 71.4 | 58.3 | 57.1 | 61.0 | 0.648 |
| Diabetes patient rate (%) | 7.1* | 41.7 | 42.9 | 30.5 | 0.029 |
| Use rate of steroid (%) | 7.1 | 21.7 | 9.5 | 13.6 | 0.360 |
| Scoliosis rate (%) | 71.4 | 50.0 | 76.2 | 64.4 | 0.154 |
| Bridging rate (%) | 78.6 | 66.7 | 76.2 | 72.9 | 0.667 |
| DISH rate (%) | 42.9* | 8.3 | 9.5 | 16.9 | 0.022 |
| Mortality rate (%) | 21.4 | 25.0 | 28.6 | 25.4 | 0.890 |

**Notes.**
*Statistical significance is $p < 0.05$.

was lower (7.1%) compared to the other groups (B: 41.7%, C: 42.9%). The steroid usage did not significantly influence spine infection rates compared to other types of infections (A: 7.1%, B: 21.7%, C: 9.5%). The overall mortality rate for MSSA bacteremia was consistent across groups at 25.4% (A: 21.4%, B: 25.0%, C: 28.6%).

### Laboratory findings and inflammatory markers
Laboratory data revealed that CRP levels at Day 1 were a stronger risk factor for spinal infection than WBC levels at Day 1. The reduction in inflammatory response, measured as the ratio of Day 3 to Day 1 WBC, was marked in both spine and implant infection groups (A: 74.2 ± 19.4, C: 84.0 ± 28.0), but large standard deviations indicated no significant trends over time.

### Imaging and diagnostic findings
Over 50% of all patients exhibited scoliosis and bridging, likely due to the higher average age. However, the rate of DISH was significantly higher in spine infection cases (A: 42.9%, compared to B: 8.3%, and C: 9.5%). All patients with spine infections had thoracic but not lumbar DISH, aligning with common radiographic findings in epidemiological literature.

### Risk factors of spine infection
The multinomial univariable logistic regression and contingency table analysis detected the higher risk factor of spine infection, the absence of diabetes ($p = 0.029$), higher Day 1 CRP levels ($p = 0.014$), and the presence of DISH ($p = 0.022$) significantly differentiated the spinal infection group.

**Table 2** Results of binominal multivariable logistic regression analysis (A *vs* B, C) on significant risk factors.

| Variables | Odds ratio | Lower CI | Upper CI | P value |
|-----------|-----------|----------|----------|---------|
| DISH | 41.750 | 1.856 | 939.023 | 0.019* |
| DM | 1.201 | 1.010 | 1.428 | 0.038* |
| CRP day1 | 23.338 | 1.127 | 483.378 | 0.042* |
| WBC3/1 | 0.964 | 0.929 | 1.000 | 0.047* |
| Age | 0.919 | 0.840 | 1.005 | 0.063 |
| bridging | 13.538 | 0.282 | 650.400 | 0.187 |
| Gender | 7.830 | 0.360 | 170.449 | 0.190 |
| Scoliosis | 2.887 | 0.271 | 30.804 | 0.380 |
| CRP3/1 | 0.995 | 0.981 | 1.009 | 0.466 |
| WBC day1 | 1.000 | 1.000 | 1.000 | 0.829 |
| Steroid use | 1.189 | 0.082 | 17.292 | 0.899 |

**Notes.**
*Statistical significant ($p < 0.05$).

## Odds ratio using the binomial multivariable logistic regression analysis

The binomial multivariable logistic regression analysis was conducted to evaluate risk factors for spinal infection by comparing Group A (spinal infections) with B and C (the others) detailed in Table 2. A markedly high odds ratio was DISH (OR = 41.750, 95% confidence interval (CI) [1.856–939.023]) and elevated CRP at the onset (OR = 23.338, 95% CI [1.127–483.378]). Lower WBC 3/1 (OR = 0.964, 95% CI [0.929–1.000]) and absence of diabetes (OR = 1.201, 95% CI [1.010–1.428]) were associated with a slightly increased likelihood of spinal infection.

These findings suggest that DISH and higher CRP levels are particularly indicative of spinal infections, while diabetes and WBC count changes are less predictive.

## Additional imaging insights

Early detection of infection sites, especially during initial stages of discitis and facet joint infections in patients under septic shock or coma, was investigated using MIP-CT. This imaging technique served as a potential surrogate marker, pinpointing areas with the highest CT values indicative of both degenerative changes and infection. For instance, areas highlighted by a black arrow in Fig. 2Ab showed the highest intensity on the spine, suggesting a predisposition to degenerative scoliosis.

## DISCUSSION

The primary question this study sought to address was whether spinal infections could be effectively predicted by factors such as present illness, inflammation markers, antibiotic effects, or spinal degenerative changes due to alignment and instability. Our findings indicate that high initial CRP levels, which reflect inflammation, are strong indicators of spinal infection, more so than WBC counts or the effect of treatment measured by changes in WBC and CRP levels from Day 1 to Day 3. Notably, the spinal infection group exhibited

higher CRP levels at the start of therapy, suggesting that elevated CRP could be associated with the complexity of diagnosing spinal infections according to IDSA guidelines (*Berbari et al., 2015*), which are sometimes only confirmed after observing changes in follow-up tests over one or two weeks.

Although diabetes is commonly cited as a risk factor for spinal infection (*Mader & Lavi, 2009*; *Souli et al., 2019*), its prevalence was lower in our study's spinal infection group compared to other groups. This discrepancy might be explained by the higher incidence of comorbid conditions such as infectious endocarditis, abscesses, or implant infections in patients with diabetes, dialysis dependence, cancer, or corticosteroid use (*Souli et al., 2019*). Moreover, while DISH is frequently associated with conditions like gout, metabolic syndrome, and heart failure (*Kiss et al., 2002*), it did not correlate with diabetes in our study. Recent research suggests DISH may be more linked to altered lipid profiles than to diabetes (*Mader et al., 2021*), positioning it as a potent, independent risk factor for spinal infection in MSSA bacteremia.

A secondary consideration was the relationship between spinal alignment, instability, and infection, particularly in terms of DISH. Our results showed that DISH-related infections were primarily located in areas without spinal fusion, potentially increasing the risk of instability in adjacent spinal segments (*Katzman et al., 2017*). The pathogenesis of DISH may involve low levels of Dickkopf-1 (DKK-1) linked to severe DISH through the Wnt pathway, suggesting a connection between metabolic inflammation and bone formation (*Bieber et al., 2020*; *Niu et al., 2017*). While decreased DKK-1 expression was initially thought to confer increased resistance to infections, spinal adjacent segment instability or changes in spinal blood flow might play a more significant role in the increased risk of infections in DISH. This association underscores the need for further studies to explore the mechanism of inflammation in DISH.

Another notable observation was the tool of CT-MIP in diagnosing facet infections more easily, where areas with the highest HU values correlated with not only degenerative changes but also infection, as in previous references (*Andre et al., 2015*; *French et al., 2015*). This technique could serve as a valuable tool for identifying potential infection sites, which are often difficult to diagnose definitively and contribute to spine dysfunction, deformity, and increased mortality.

This study was limited by its small sample size of 14 spinal infection patients from a single center, which may affect the generalizability of the findings. Additionally, the categorization of patients into control and experimental groups was not feasible, limiting the application of our findings in a standard infection risk model. Future studies should expand the sample size and refine the methodological approach to validate our findings and further elucidate the risks associated with spinal infections. Furthermore, the utility of CT-based diagnostics in longitudinal infection risk assessment warrants additional investigation to confirm its practical benefits and adapt it for broader clinical use.

## CONCLUSIONS

The risk factors of spine infection with MSSA bacteremia included relatively lower rates of diabetes and higher CRP levels at onset and DISH compared to the other groups. DISH

might contribute to spinal joint instability, similar to adjacent segmental disease, and thus presents a distinct risk factor for spinal infection beyond previously recognized factors in MSSA bacteremia.

## ACKNOWLEDGEMENTS

The author would like to thank Noboru Miyagi M.D. Ph.D., Hideshi Harada M.D., and the Sapporo Tokushukai Hospital medical staff, especially the Department of Orthopedics, Emergency Center, Primary Care Center, Radiology, Internal Medicine and Naoki Wada (Clinical Laboratory Center). The author is also grateful to Kazuhito Hitachi and Taketo Iwase in Sapporo Minami Orthopedics Hospital and Hokkaido University Hospital Clinical Research and Medical Innovation Center.

### Funding

The authors received no funding for this work.

### Competing Interests

The authors declare there are no competing interests.

### Author Contributions

- Kaori Endo conceived and designed the experiments, performed the experiments, analyzed the data, prepared figures and/or tables, authored or reviewed drafts of the article, and approved the final draft.

### Human Ethics

The following information was supplied relating to ethical approvals (i.e., approving body and any reference numbers):

The Tokushukai Group Ethics Committee granted Ethical approval to carry out the study within its facilities .

### Data Availability

The raw measurements are available in the Supplementary Files.

### Supplemental Information

Supplemental information for this article can be found online at http://dx.doi.org/10.7717/peerj.18432#supplemental-information.

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
