# Peer review of "Diffuse idiopathic skeletal hyperostosis was the specific risk factors of methicillin-susceptible Staphylococcus aureus spine infection: a retrospective study in a single center"

_PeerJ, doi:10.7717/peerj.18432_

## Round 0.1 · original submission · Minor Revisions

Dear Authors,

Thank you for submitting your work to us. We have received feedback from our peers that suggests major revisions are needed. Please review these points carefully and provide us with your responses. Particularly, Reviewer 2 has highlighted some thoughts that could significantly strengthen the manuscript and should be included. The methods and statistical sections need to be improved per reviewer 2.

We look forward to receiving your revised submission.

Best wishes,
Dr. Nagendran Tharmalingam.

Reviewer 1 ·

Basic reporting

Perhaps you can clearly state the study's purpose in relation to your hypothesis on DISH. The hypothesis was stated clearly, but your purpose of the study was very general; it could be more specific in relation to the title of your study.

Experimental design

no comment

Validity of the findings

The discussion could describe in detail, concerning the title of the article, why DISH was the specific risk factor for MSSA spine infection and how it was related to it. It was described in the article but wasn't highlighted as to why; perhaps, it could do so more clearly.

Annotated reviews are not available for download in order to protect the identity of reviewers who chose to remain anonymous.

Reviewer 2 ·

Basic reporting

Please avoid using abbreviations in the title.

Experimental design

Overall, please expand the Methods section with more details. For example,
1. In Subjects, define more clearly the inclusion-exclusion criteria.
2. In Data Analysis, I don't see Figure A/Ab. The additional question to be investigated should be stated earlier. Also, for "The effectiveness of the antibiotic treatment was assessed by comparing the rates of WBC and CRP" - which method is used for the comparison? It is mentioned in the results that the comparison was significant, how did you determine the significance?
3. In Statistical analysis, please include all details about modeling as mentioned in the Results. For example, how did you measure multicollinearity? How did you develop the more accurate prediction model? Why would you say its more "accurate"? Which variables were included in the model?

In addition, how did you handle potential confounding in the modeling? How did you choose your patient factors?

Validity of the findings

No comment

Additional comments

No comment

---

## Round 0.2 · Minor Revisions

Dear Authors,

Please address reviewer 2's comment on Table 1 to send the final decision.

Thank you.
Dr. Nagendran Tharmalingam

Reviewer 1 ·

Basic reporting

It is clear now with relevant results to the hypotheses

Experimental design

The research question well defined and methods well described

Validity of the findings

Data in results are clear and conclusion are well stated.

Reviewer 2 ·

Basic reporting

None

Experimental design

Thank you for addressing the comments. No further comments here.

Validity of the findings

None.

Additional comments

Perhaps the layout of table 1 could be clearer: a more detailed table title & more specific description of the numbers in the table.

---

## Round 0.3 · accepted · Accept

Dear Authors,

Thank you for submitting your critical work to us. We would like to inform you that your work has been accepted for publication. The production house may contact you regarding any typeset questions they may have.

Best wishes for your future submissions to us.

Kindly,
Dr. Nagendran Tharmalingam
Handling Editor

Reviewer 2 ·

Basic reporting

no comment

Experimental design

no comment

Validity of the findings

no comment

Additional comments

For better formatting, last two sentences in the title of table 1 can be moved to footnote.